materials science

Cu(InGa)Se2, quaternary target, Se-free annealing, Se-containing annealing

**Author for correspondence:**
Leng Zhang
e-mail: zhanglengxixi@163.com

This article has been edited by the Royal Society of Chemistry, including the commissioning, peer review process and editorial aspects up to the point of acceptance.

# Effects of annealing atmosphere on the performance of Cu(InGa)Se2 films sputtered from quaternary targets

## Leng Zhang[1], Yongyi Yu[1], Jing Yu[1] and Yaowei Wei[2]

[1]School of Electronics and Information Engineering, Jinling Institute of Technology, 211169 Nanjing, Jiangsu, People's Republic of China
[2]State Center for International Cooperation on Designer Low-carbon and Environmental Materials, School of Materials Science and Engineering, Zhengzhou University, 450001 Zhengzhou, People's Republic of China

LZ, 0000-0002-6453-4111

Quaternary sputtering without additional selenization is a low-cost alternative method for the preparation of Cu(InGa)Se2 (CIGS) thin film for photovoltaics. However, without selenization, the device efficiency is much lower than that with selenization. To comprehensively examine this problem, we compared the morphologies, depth profiles, compositions, electrical properties and recombination mechanism of the absorbers fabricated with and without additional selenization. The results revealed that the amount of surface Se on CIGS films annealed in a Se-free atmosphere is less than that on CIGS films annealed in a Se-containing atmosphere. Additionally, the lower amount of surface Se reduced the carrier concentration, enhanced the resistivity of the CIGS film and allowed CIGS/ CdS interface recombination to be the dominant recombination mechanism of CIGS device. The increase of interface recombination reduced the efficiency of the device annealed in a Se-free atmosphere.

## 1. Introduction

In the industrial production of Cu(InGa)Se2 (CIGS) photovoltaic devices, absorbers are usually produced by a two-step process comprising sputter deposition of a Cu-In-Ga alloy precursor, followed by post-selenization and sulfurization [1,2]. Sputtering has great advantages when the technology is transferred from laboratory-scale solar cells to production-scale panels, because it produces large-area film homogeneity [3–5]. Another promising

method is based on the sputtering of CIGS quaternary targets and post-annealing; it involves high materials usage and less reliance on toxic selenium powder or H2Se [6,7]. The post-annealing in this method includes either a Se-containing or a Se-free atmosphere treatment. The Se-free atmosphere annealing has more potential because it completely avoids the toxic selenium powder or H2Se. Various Se-free fabrication routines have been reported [8–10]. For example, Frantz *et al.* [5] and Chen *et al.* [6] obtained CIGS device conversion efficiencies of 8%–10% by using one-step sputtering at high substrate temperatures with no post-treatment. However, the efficiency of a solar cell prepared in a Se-free atmosphere is far less than those prepared in a Se-containing annealing atmosphere. It is unclear why CIGS device efficiency is lower when prepared by Se-free annealing. Franz *et al.* found that the upper limit of the device efficiency is most likely the result of impurities in the sputtering target, which leads to electronic defects in the absorber [5]. Park *et al.* reported that the formation of Se vacancies ($V_{se}$) on a CIGS film surface during Se-free treatment is the main limiting factor of the device efficiency [9].

The quality of the absorber, the contact between the absorber and the back electrode, and the interfacial matching between the buffer and absorber layers are three important factors that determine the conversion efficiency of the CIGS solar cell [10]. Previously, we reported that for Mo–CIGS interfaces, the contact is ohmic when the devices are annealed in either a Se-free atmosphere or a Se-containing atmosphere [11]. Furthermore, the phase structures of CIGS films annealed in Se-free and Se-containing atmospheres are both chalcopyrite with nearly identical diffraction peaks [11]. Thus, the limiting factor of the low efficiency in devices made in a Se-free atmosphere is attributed to the quality of the absorber and CIGS–CdS interfacial matching. To explore the effect of annealing atmosphere on the device performance, the morphologies, elemental depth profiles, quantitative compositions, electrical properties, and recombination mechanisms of the device are analysed.

## 2. Experimental details

The base pressure and the working argon pressure for CIGS deposition were $2.0 \times 10^{-3}$ Pa and 0.7 Pa, respectively. The CIGS films were deposited by sputtering from a quaternary CIGS target at room temperature in a pure argon discharge atmosphere at a middle frequency of power density approximately 30 W cm$^{-2}$. The CIGS precursors were then annealed in either a Se-free or a Se-containing atmosphere. Some of the fabricated glass/Mo/CIGS samples were placed in a quartz tube furnace that was pumped to a base pressure of $2.0 \times 10^{-3}$ Pa and filled with nitrogen at 0.5 atm. The annealing was performed at 550°C for 40 min. The heating rate was 15°C min$^{-1}$ and the samples were allowed to cool naturally. For comparison, some of the fabricated glass/Mo/CIGS samples were placed in a furnace for selenization at 550°C for 40 min in atmosphere filled with hydrogen selenide and argon (H2Se/Ar). The pressure was approximately 10 000 Pa with a H2Se/Ar flux ratio of 1 : 100.

The morphologies of the films were imaged with a field-emission scanning electron microscope (SEM, Ziess Sigma) and the elemental composition was analysed with energy-dispersive X-ray spectroscopy (EDS) system in the SEM. The accelerating voltage to collect EDS was 15 kV. The depth distributions were measured by secondary ion mass spectroscopy (SIMS, ION-TOF GmbH instrument, TOF.SIMS 5-100), and the sputtering was provided by bismuth primary ion bombardment at 30 keV. The electrical properties were characterized by Hall measurement (Hall, HL5500PC, Nanometric). A device structure of Mo/CIGS/CdS/i-ZnO/AZO/Ni-Al was used in this research. The fabrication details of CIGS solar cell have been described in our previous literature [11]. The temperature-dependent current–voltage (J–V) measurements were conducted under AM1.5 (100 mW cm$^{-2}$) illumination using a solar simulator to examine the recombination mechanisms.

## 3. Results

The surface and cross-sectional morphologies of CIGS films annealed in Se-free and Se-containing atmospheres are shown in figure 1*a* and *b*, respectively. In figure 1*a*, the surface grains are distributed in a cauliflower shape, and the grain boundaries are round. Additionally, the film has a flat and compact morphology, and the arrangement of grains is not dense because there are many holes between the grains. In figure 1*b*, the grains in the cross-sectional view are approximately 1 µm, with close contact among grains. Furthermore, the grain boundaries in the surface region in figure 1*b* are straight and the arrangement of the grains is denser with fewer holes between grains than that in figure 1*a*.

According to the principle of chemical equilibrium movement, gaseous H2Se can facilitate the transition of the film from an amorphous state to a crystalline state. Overall, Se in the atmosphere

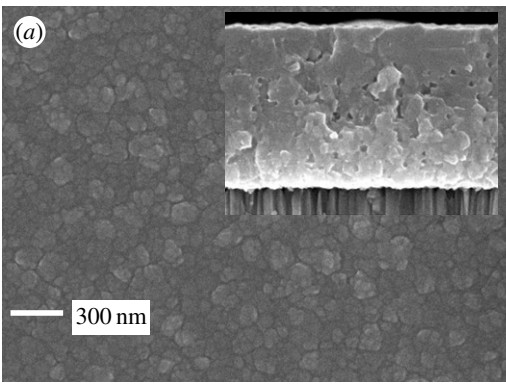
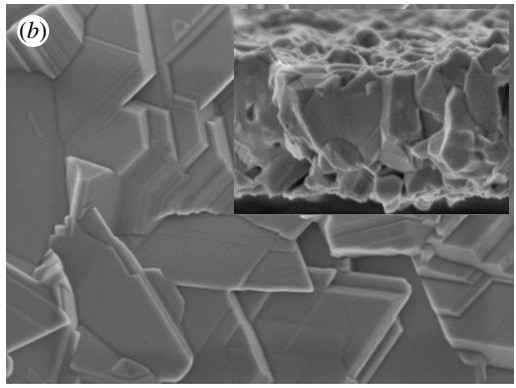

**Figure 1.** (*a*) Plane view SEM micrographs of a CIGS films annealed in a Se-free atmosphere. The inset is a cross-sectional view. (*b*) Plane view SEM micrographs of a CIGS films annealed in a Se-containing atmosphere. The inset is a cross-sectional view.

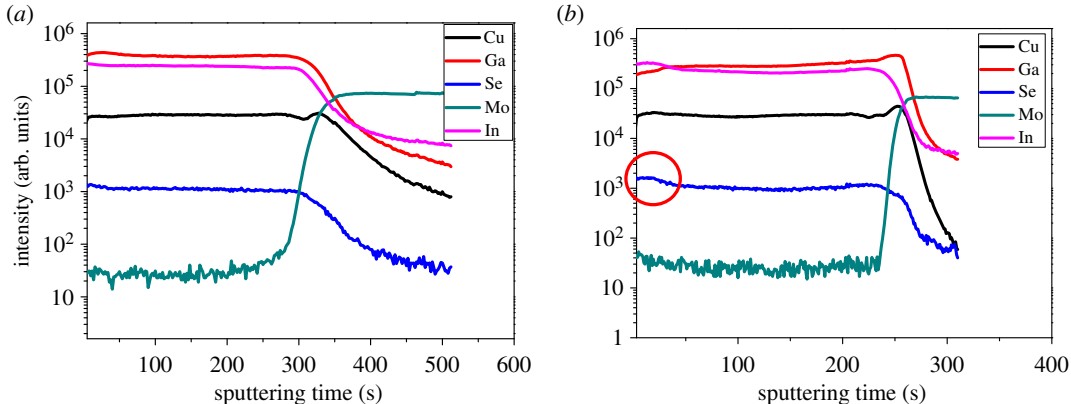

**Figure 2.** (*a*) Element distribution profile in CIGS film acquired with SIMS prepared via Se-free annealing; (*b*) element distribution profile in CIGS film acquired with SIMS prepared via Se-containing annealing.

accelerates the chemical reactions and promotes atomic diffusion. It has been reported that the grain size has little effect on the properties of CIGS solar cells when the conversion efficiency is below 16% [12–14]. Therefore, the small grain size of the CIGS solar cells made in a Se-free atmosphere may not be the key factor in poor device performance.

SIMS measurements are used to analyse the elemental depth profile of the CIGS films annealed in Se-free and Se-containing atmospheres; typical results are shown in figure 2. The thicknesses of the two films are almost the same, but the etching rates are slightly different; therefore, the sputtering times have slight differences. Overall, the elemental depth profile of the two samples is similar, but there are slight differences, as discussed below.

The indium and gallium depth profiles of the CIGS films annealed in a Se-free atmosphere are almost parallel to the horizontal axis, which indicates that the indium and gallium levels, and the quantity Ga/(Ga+In) (GGI), are almost constant with depth. Thus, the CIGS band gap is almost constant with depth because it correlates with GGI [7]. However, the gallium concentration at the surface of the CIGS film annealed in a Se-containing atmosphere is lower than that annealed in a Se-free atmosphere. The gallium proportion increases gradually with depth for the film annealed in a Se-containing atmosphere. At the junction of the Mo and CIGS layers, the gallium concentration increases, which indicates gallium aggregates in this region. This is because of the lower reaction rate between gallium and selenium relative to that between indium and selenium. At the CIGS film surface, the reaction rate of indium, copper and hydrogen selenide is faster than that of gallium, copper and hydrogen selenide, which 'drives' gallium to the bottom of the CIGS film. The increase in GGI with depth has positive and negative consequences [15]. On the one hand, in the CIGS solar cell band gap diagram, when GGI increases, the conduction band bends upward and electrons in the neutral zone are subjected to an additional electric field. This is beneficial to electron transport from the neutral zone

**Table 1.** Surface composition via EDS for CIGS annealed in Se-free and Se-containing atmosphere.

| annealing method | Cu (at%) | In (at%) | Ga (at%) | Se (at%) | GGI | Se/Metal |
|---|---|---|---|---|---|---|
| Se-containing annealing | 28.01 | 19.48 | 5.53 | 47.18 | 0.22 | 0.89 |
| Se-free annealing | 28.22 | 19.48 | 6.13 | 46.17 | 0.24 | 0.86 |

**Table 2.** Electrical property parameters of CIGS films annealed in Se-free and Se-containing atmosphere.

| annealing condition | conductivity type | carrier mobility ($cm^2 \cdot V^{-1} \cdot s^{-1}$) | carrier concentration ($cm^{-3}$) | resistivity ($\Omega$ cm) |
|---|---|---|---|---|
| Se-containing | p | 4.10 | $2.20 \times 10^{17}$ | 6.92 |
| Se-free | p | 9.98 | $5.21 \times 10^{16}$ | 11.99 |

to the space-charge region. On the other hand, the open-circuit voltage ($V_{OC}$) of the solar cell is closely correlated with the band gap at the CIGS film surface. The low surface-gallium concentration creates a low $V_{OC}$. Combining these two aspects of the solar cell, one can conclude that the quantity GGI with depth in the CIGS film is not the key factor limiting the performance of CIGS solar cells prepared by annealing in a Se-free atmosphere.

Furthermore, the selenium depth profile stays almost parallel to the horizontal axis for the Se-free annealed sample, while it bends upward (circled in red) on the surface of CIGS film prepared via Se-containing annealing.

A more quantitative composition analysis was conducted with EDS in the SEM. By adjusting the acceleration voltage of the electrons, the measurement depth can be as much as 400 nm. Therefore, the EDS results reflect the composition of the CIGS films in the upper 400 nm and can be used to compare different samples. Typical compositions of the CIGS films are listed in table 1. For the film annealed in the Se-containing atmosphere, the GGI is approximately 0.22, while it is approximately 0.24 for the film annealed in the Se-free atmosphere, which is in accordance with the SIMS results. However, the ratio of Se/metal is approximately 0.89 for the film annealed in the Se-containing atmosphere, while that for the film annealed in the Se-free atmosphere is approximately 0.86. This indicates that the Se-containing atmosphere adds selenium into CIGS films, which could suppress the formation of donor defects of $V_{Se}$. On the other hand, for the Se-free atmosphere, the Se deficiency at the film surface creates donor defects $V_{Se}$, which may compensate for the hole concentration and increase defect state density at the interface.

Hall measurements were used to analyse the electrical properties of the CIGS films prepared by Se-free and Se-containing annealing atmospheres. The results are listed in table 2. The conductivity type, carrier mobility and carrier concentration of the two films basically meet the electrical performance requirements. Generally, the resistivity decreases with doping concentration. The CIGS film annealed in the Se-containing atmosphere has lower resistivity because the doping concentration is higher. In the Se-free condition, there are many Se vacancies in the film, especially on the surface, which neutralize the hole concentration produced by the copper vacancies. Therefore, the carrier concentration in the CIGS film annealed in the Se-free atmosphere is lower.

The device performance, especially the $V_{OC}$, is affected by recombination mechanisms. To discuss the mechanisms, temperature-dependent J–V curves are shown in figure 3a. The $V_{OC}$ of the device is around 530 mV at 298 K. If $V_{OC}$ at 0 K, obtained by extrapolating the linear fit in figure 3b, is less than $E_g/q$, where $E_g$ is the band gap of CIGS absorption layer, the loss of conversion efficiency is g attributed to interface recombination [16]. As shown in figure 3b, the $V_{OC}$ of the solar cell at 0 K is estimated to be approximately 1.03 V. With an $E_g$ estimated to be 1.15 eV from our previous research, this suggests that interface recombination is the dominant recombination mechanism. Previously, we found that the $V_{OC}$ at 0 K of the solar cell annealed in the Se-containing atmosphere was higher than that annealed in Se-free atmosphere [17]. Because the concentration of Se at the top surface of the absorber annealed in the Se-containing atmosphere is higher than that for the absorber annealed in the Se-free atmosphere, it is likely that the reduced performance of CIGS solar cell annealed in the Se-free atmosphere is attributed to the decreased selenium in the surface layer. Hence the efficiency of

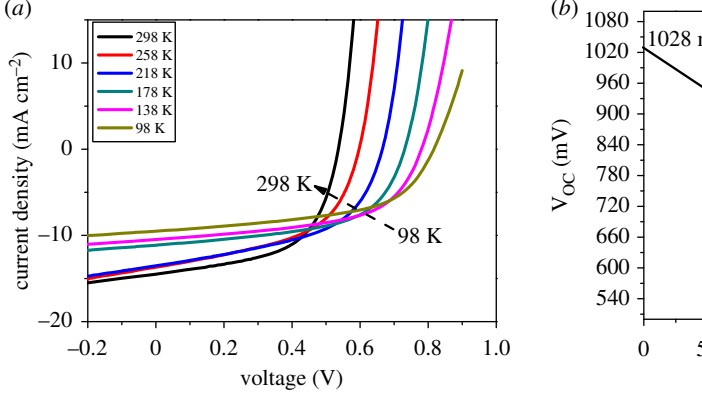
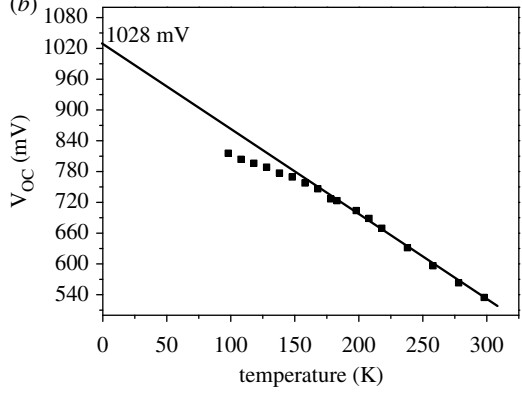

**Figure 3.** (*a*) Temperature dependence of the J–V curve of the device annealed in a Se-free atmosphere. (*b*) Fitting analysis of $V_{OC}$ with temperature.

the CIGS solar cell annealed in Se-free atmosphere could be improved by using excessive selenium in the target.

## 4. Conclusion

The effects of annealing atmosphere on the performance of CIGS films were comprehensively investigated. In summary, the Se-free annealing procedure created CIGS thin films that exhibit uniform elemental distributions without Ga segregation at the bottom of the film. The Se-free annealing creates Se vacancies, which lead to a dominant recombination mechanism at the absorber–buffer interface. From the depth profiles, the composition of the surface of CIGS film and the temperature-dependent J–V curves, it is likely that the key limiting factor for the efficiency of the device annealed in a Se-free atmosphere is the low concentration of selenium in the film surface. Sufficient selenium in the CIGS target is thus expected to improve the device efficiency for CIGS films annealed in a Se-free atmosphere.

Data accessibility. All raw data, code, analysis files and materials associated with this study are deposited at Dryad: https://doi.org/10.5061/dryad.00000001j. The datasets supporting this article are also included in the electronic supplementary material.

Authors' contributions. L.Z. designed the study and wrote the manuscript. J.Y. and Y.Y. analysed the data. Y.W. revised the draft for important intellectual content. All authors gave final approval for publication.

Competing interests. We declare we have no competing interests.

Funding. This work was supported by National Natural Science Foundation of China (61904071) and Jinling Institute of Technology High-level Talent Research Startup Fund ( jit-b-201822).

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
