## [Reviewer comments · Royal Society Open Science]

Review History

RSOS-200662.R0 (Original submission)

Review form: Reviewer 1

Is the manuscript scientifically sound in its present form?

Yes

Are the interpretations and conclusions justified by the results?

Yes

Is the language acceptable?

Yes

Do you have any ethical concerns with this paper?

No

Have you any concerns about statistical analyses in this paper?

No

Recommendation?

Accept with minor revision (please list in comments)

Comments to the Author(s)

The authors evaluated the influence of annealing atmosphere on the photovoltaic properties by measuring the properties of the Cu(InGa)Se₂ such as the morphology, the depth profile, the quantitative composition, the electrical properties and the recombination mechanisms. The authors found that the selenization increases the surficial Se of CIGS film, which increases the carrier concentration, reduces the resistivity, and reduces the interface recombination. The manuscript records some new results based on other researchers' results in the field. The results are adequately presented in light of the existing literature. Besides, the data in the dryad could support the argument of the manuscript.

I recommend the acceptance of the manuscript with a minor revision addressing the following comments.

1. The language of the manuscript needs some polish.
2. The style of the figure caption should be uniform.
3. Is the distribution of Se uniform on the surface? Was the film surface etched?
4. In the second paragraph of introduction part, the authors claim that the interface of CdS-CIGS are one of the three important factors that determine the solar cell performance. How to evaluate the interface quality of CdS-CIGS except the device recombination mechanism?
5. How does the Se-containing annealing atmosphere influence the defects of VSe? How does VSe influence the device performance?
6. Are the energy band diagrams of the films prepared with and without selenium atmosphere necessary, so as to make a direct comparison and observation?
7. Can the author explain why indium, copper and hydrogen selenium react faster than gallium, copper and hydrogen selenium?

Review form: Reviewer 2

Is the manuscript scientifically sound in its present form?

Yes

Are the interpretations and conclusions justified by the results?

No

Is the language acceptable?

Yes

Do you have any ethical concerns with this paper?

No

Have you any concerns about statistical analyses in this paper?

No

Recommendation?

Major revision is needed (please make suggestions in comments)

Comments to the Author(s)

The authors performed a technologically-relevant study on the importance of annealing in a Se environment after sputtering Cu(In,Ga)Se₂ films from a quaternary target. In order for the authors to build the case for their conclusions more convincingly, I humbly suggest that they consider the following points.

1. What accelerating voltage did they use to collect EDS?
2. How did they form contacts for Hall measurements?

3. What were the device stacks?
4. They calculated band gap using the films' overall composition, but they only reported the films' surface composition by EDS. The measured overall compositions, or how they were estimated, should be reported.
5. Their study compared Se vs Se-free annealing, so why did they not include JV data for both sample types?
6. They described in detail the small change in surface Ga/(Ga+In) composition, but they did not discuss its effects on the devices. How do they know that the surface Ga/(Ga+In) composition did not dominate the 'superficial Se' effect?

Decision letter (RSOS-200662.R0)

Dear Dr Zhang:

Title: Influence of annealing atmosphere on performances of CIGS film by sputtering from quaternary targets
Manuscript ID: RSOS-200662

The editor assigned to your manuscript has now received comments from reviewers. We would like you to revise your paper in accordance with the referee and Subject Editor suggestions which can be found below (not including confidential reports to the Editor). Please note this decision does not guarantee eventual acceptance.

Please submit your revised paper before 02-Aug-2020. Please note that the revision deadline will expire at 00.00am on this date. If we do not hear from you within this time then it will be assumed that the paper has been withdrawn. In exceptional circumstances, extensions may be possible if agreed with the Editorial Office in advance. We do not allow multiple rounds of revision so we urge you to make every effort to fully address all of the comments at this stage. If deemed necessary by the Editors, your manuscript will be sent back to one or more of the original reviewers for assessment. If the original reviewers are not available we may invite new reviewers.

On behalf of the Subject Editor Professor Anthony Stace and the Associate Editor Dr Dattatray Late.

RSC Associate Editor:
Comments to the Author:
(There are no comments.)

RSC Subject Editor:
Comments to the Author:
(There are no comments.)

Reviewers' Comments to Author:
Reviewer: 1

Comments to the Author(s)

The authors evaluated the influence of annealing atmosphere on the photovoltaic properties by measuring the properties of the Cu(InGa)Se₂ such as the morphology, the depth profile, the quantitative composition, the electrical properties and the recombination mechanisms. The authors found that the selenization increases the surficial Se of CIGS film, which increases the carrier concentration, reduces the resistivity, and reduces the interface recombination. The manuscript records some new results based on other researchers' results in the field. The results are adequately presented in light of the existing literature. Besides, the data in the dryad could support the argument of the manuscript.

I recommend the acceptance of the manuscript with a minor revision addressing the following comments.

1. The language of the manuscript needs some polish.
2. The style of the figure caption should be uniform.
3. Is the distribution of Se uniform on the surface? Was the film surface etched?
4. In the second paragraph of introduction part, the authors claim that the interface of CdS-CIGS are one of the three important factors that determine the solar cell performance. How to evaluate the interface quality of CdS-CIGS except the device recombination mechanism?
5. How does the Se-containing annealing atmosphere influence the defects of VSe? How does VSe influence the device performance?
6. Are the energy band diagrams of the films prepared with and without selenium atmosphere necessary, so as to make a direct comparison and observation?
7. Can the author explain why indium, copper and hydrogen selenium react faster than gallium, copper and hydrogen selenium?

Reviewer: 2

Comments to the Author(s)

The authors performed a technologically-relevant study on the importance of annealing in a Se environment after sputtering Cu(In,Ga)Se₂ films from a quaternary target. In order for the authors to build the case for their conclusions more convincingly, I humbly suggest that they consider the following points.

1. What accelerating voltage did they use to collect EDS?
2. How did they form contacts for Hall measurements?
3. What were the device stacks?
4. They calculated band gap using the films' overall composition, but they only reported the films' surface composition by EDS. The measured overall compositions, or how they were estimated, should be reported.
5. Their study compared Se vs Se-free annealing, so why did they not include JV data for both sample types?
6. They described in detail the small change in surface Ga/(Ga+In) composition, but they did not discuss its effects on the devices. How do they know that the surface Ga/(Ga+In) composition did not dominate the 'superficial Se' effect?

Author's Response to Decision Letter for (RSOS-200662.R0)

See Appendix A.

Decision letter (RSOS-200662.R1)

Dear Dr Zhang:

Title: Effects of annealing atmosphere on the performance of Cu(InGa)Se₂ films sputtered from quaternary targets
Manuscript ID: RSOS-200662.R1

It is a pleasure to accept your manuscript in its current form for publication in Royal Society Open Science. The chemistry content of Royal Society Open Science is published in collaboration with the Royal Society of Chemistry.

On behalf of the Subject Editor Professor Anthony Stace and the Associate Editor Dr Dattatray
Late.

RSC Associate Editor
Comments to the Author:
Accept

Reviewer(s)' Comments to Author:

Appendix A

Dear Editors and Reviewers,

Thank you for your letter and for the reviewers' comments concerning our manuscript entitled "Effects of annealing atmosphere on the performance of Cu(InGa)Se₂ films sputtered from quaternary targets". Those comments are all valuable and very helpful for revising and improving our paper, as well as the important guiding significance to our researches. We have studied comments carefully and have made correction which we hope to meet with approval. Revised portion are marked in red in the paper. The responds to the reviewer's comments are as flowing:

Reviewers' Comments to Author:

Reviewer: 1

Comments to the Author(s)

The authors evaluated the influence of annealing atmosphere on the photovoltaic properties by measuring the properties of the Cu(InGa)Se₂ such as the morphology, the depth profile, the quantitative composition, the electrical properties and the recombination mechanisms. The authors found that the selenization increases the surficial Se of CIGS film, which increases the carrier concentration, reduces the resistivity, and reduces the interface recombination. The manuscript records some new results based on other researchers' results in the field. The results are adequately presented in light of the existing literature. Besides, the data in the dryad could support the argument of the manuscript.

I recommend the acceptance of the manuscript with a minor revision addressing the following comments.

1. The language of the manuscript needs some polish.

We are very sorry for our language problems. We have revised and improved the English writing of the whole manuscript. We tried our best to improve the manuscript and made many changes in the manuscript. These changes will not influence the content and framework of the paper. The marked revision details are described in revised manuscript with marked changes.

2. The style of the figure caption should be uniform.

Thank you for your valuable advice. We have re-written the caption of Fig.2 as follows:

Fig.2 (a)Element distribution profile in CIGS film acquered with SIMS prepared via Se-free annealing;
(b) Element distribution profile in CIGS film acquered with SIMS prepared via Se-containing annealing.

3. Is the distribution of Se uniform on the surface? Was the film surface etched?

As is known, sputtering method has greater advantage on the film uniformity than evaporation method and solution method. The sputtering method based on quaternary CIGS target has greater advantage on uniformity than sputtering method based on alloy target, as the latter method involve more chemical reaction and atomic migrations. Besides, several points of the film surface were measured via EDS, which got similar Se content. Before sampling, the films were performed a simple surface cleaning by argon ion, which means the film surface was etched for a little while.

4. In the second paragraph of introduction part, the authors claim that the interface of CdS-CIGS are one of the three important factors that determine the solar cell performance. How to evaluate the interface quality of CdS-CIGS except the device recombination mechanism?

Thank you for your thoughtful comments. The CdS-CIGS heterojunction interface has amount of lattice mismatch, leading to a high interface state density. The interface state density determines the

interface quality. By analyzing the device recombination mechanism, the integral interface quality could be evaluated indirectly. Also, by observing the atomic arrangement of the interface under high-power transmission electron microscope, the interface quality could be evaluated directly; however, this method could only be used to analyze partial interface quality.

Some of our experience can be used to evaluate the interface quality. CIGS films with large surface fluctuations are not conducive to good interface quality. The pores on the surface will form bubbles when depositing CdS and prevent the liquid from penetrating to the surface of the absorption layer during the deposition process. Or due to surface tension, it is difficult for the liquid to infiltrate the elongated pores, which hinders the deposition of CdS layer and CdS layer cannot cover the absorption layer completely and uniformly, leading to poor interface quality.

5. How does the Se-containing annealing atmosphere influence the defects of V_{Se} ? How does V_{Se} influence the device performance?

Thanks of your good advice. We have revised the corresponding description as follows: This indicates that the Se-containing atmosphere adds selenium into CIGS films, which could suppress the formation of donor defects of V_{Se} . The Se deficiency at the film surface creates donor defects V_{Se} , which may compensate for the hole concentration and increase defect state density at the interface.

6. Are the energy band diagrams of the films prepared with and without selenium atmosphere necessary, so as to make a direct comparison and observation?

Thanks of your good advice. The energy band diagrams of these two films could reflect the variation of the conduction band along depth, so that the movement of electrons can be reflected more intuitively. From the accurate element distribution profile, the energy band diagrams of the films could be drawn.

However, in this research, the element distribution profile was obtained by SIMS. SIMS is used for qualitative analysis, not suitable for quantitative analysis. Therefore there are some difficulties in drawing the energy band diagram. We wish you could understand our difficulties.

7. Can the author explain why indium, copper and hydrogen selenium react faster than gallium, copper and hydrogen selenium?

This is determined by the kinetic parameters of the chemical reaction. Similar phenomenon were also observed in the fabrication method of alloy target sputtering with selenization, such as K. Kushia, Current Status and Future Prospects of Solar Frontier K.K., presented at the IW-CIGSTech 5, Berlin, Germany, April 2-3, 2014.

Indium, copper and hydrogen selenium react faster than gallium, copper and hydrogen selenium, leading to the gallium aggregation at the bottom of CIGS film. In order to get the U or $\sqrt{\quad}$ -shaped graded band gap, the sulfuration followed with selenization is necessary in Solar Frontier and many other researchers' work.

Reviewer: 2

Comments to the Author(s)

The authors performed a technologically-relevant study on the importance of annealing in a Se environment after sputtering Cu(In,Ga)Se₂ films from a quaternary target. In order for the authors to build the case for their conclusions more convincingly, I humbly suggest that they consider the

following points.

1. What accelerating voltage did they use to collect EDS?

In this research, the accelerating voltage to collect EDS was 15kV. Relevant description has been added in paragraph 3, page 2.

2. How did they form contacts for Hall measurements?

The electrical properties were characterized by Hall measurement (Hall, HL5500PC, Nanometric). The strength of the applied magnetic field is 0.514 T, and the applied voltage is 20 mV. Generally the size of the measured sample is 1 cm×1 cm, and the thickness of CIGS film is 1~2 μm. The CIGS film is deposited on the glass directly without depositing Mo layer. The four probes of Hall system were in close contact with the four corners of the square sample. When there is a large contact resistance between the probe and the sample, a liquid In-Ga alloy should be used to reduce the contact resistance.

3. What were the device stacks?

Thanks of your careful work. A device structure of Mo/CIGS/CdS/i-ZnO/AZO/Ni-Al was used in this research. The fabrication of CIGS solar cell could be referred to reference 11. Relevant description has been supplemented in paragraph 3, page 2.

4. They calculated band gap using the films' overall composition, but they only reported the films' surface composition by EDS. The measured overall compositions, or how they were estimated, should be reported.

As is known, the bandgap has dependence on the Ga/(Ga+In) (GGI). With GGI increasing from 0 to 1, the bandgap of CIGS film increases from 1.06 eV to 1.68 eV. For the U-shaped graded bandgap structure in CIGS absorber, the bandgap is calculated when GGI reaches the lowest.

In this research, the bandgap gradient is not very large, for the Se-free annealing film, GGI is almost constant along with CIGS depth. The gallium concentration at the surface of CIGS film annealed in Se-containing atmosphere is lower than that annealed in Se-free atmosphere, which means that the latter film has wider bandgap.

The bandgaps of the CIGS absorbers annealed in Se-containing atmosphere derived from the EQE curves is 1.15 eV in our previous work (reference 17), which suggests the bandgap of CIGS absorbers annealed in Se-containing atmosphere is higher than 1.15 eV.

Relevant description has been supplemented in paragraph 2, page 4.

5. Their study compared Se vs Se-free annealing, so why did they not include JV data for both sample types?

Fig.1 the JV data for both sample types

The JV data for both sample types was obtained in Fig.1. The figure wasn't included in the manuscript as the conclusion that the device efficiency without selenization is much lower than that with selenization had been emphasized in the abstract part and in the introduction part.

6. They described in detail the small change in surface Ga/(Ga+In) composition, but they did not discuss its effects on the devices. How do they know that the surface Ga/(Ga+In) composition did not dominate the 'superficial Se' effect?

Thank you for your careful work. As is known, Ga and In have the same valence of +3. Under normal circumstances, the change of Ga/(Ga+In) has little effects on the distribution of Se. in addition, for the film annealed in Se-containing atmosphere, the gallium aggregates at the bottom of CIGS film; Meanwhile, selenium maintains constant at the bottom of CIGS film. Therefore, it can be concluded that the surface Ga/(Ga+In) composition did not dominate the 'superficial Se' effect.

Special thanks to you for your good comments.

We appreciate for Editors/Reviewers' warm work earnestly, and hope that the correction will meet with approval.

Once again, thank you very much for your comments and suggestions.